

# Neutrino experiments probe hadrophilic light dark matter

**Yohei Ema[1], Filippo Sala[2] and Ryosuke Sato[1]**

**1** DESY, Notkestraße 85, D-22607 Hamburg, Germany
**2** LPTHE, CNRS & Sorbonne Université, 4 Place Jussieu, F-75252 Paris, France

## Abstract

We use Super-K data to place new strong limits on interactions of sub-GeV Dark Matter (DM) with nuclei, that rely on the DM flux inevitably induced by cosmic-ray upscatterings. We derive analogous sensitivities at Hyper-K and DUNE and compare them with others, e.g. at JUNO. Using simplified models, we find that our proposal tests genuinely new parameter space, allowed both by theoretical consistency and by other direct detection experiments, cosmology, meson decays and our recast of monojet. Our results thus motivate and shape a new physics case for any large volume detector sensitive to nuclear recoils.

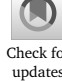

# 1 Introduction

Dark Matter (DM) is perhaps the strongest evidence for physics beyond the Standard Model (BSM), and thus motivates an intense experimental effort to determine its still elusive properties. Direct detection [1] (DD) experiments constitute one of the leading avenues to probe the non-gravitational interactions of DM particles, and have so far mostly focused on a DM mass range roughly between a GeV and tens of TeV. These searches are now in a mature stage, and no DM signal has been detected so far [2–4]. In addition, the LHC has not found clear signs of the BSM physics that, independently of DM, motivates that mass range [5]. Therefore, in the past few years, the community has investigated more in depth other DM mass regimes, and in particular the one of sub-GeV DM (see e.g. the report [6]).

Direct detection experiments rely on the scattering of DM with a target particle on the Earth, whose effects, like the recoil energy of the target, can be detected. The average velocity of DM particles in the solar neighbourhood, $v \approx 10^{-3}$, implies that for any DM mass there exists an upper limit on the final kinetic energy of the target $T$, $k_{\max} = 2v^2\mu_{\mathrm{DM}T}^2/m_T \times (1 + O(v^2))$, where $m_T$ is the target mass and $\mu_{\mathrm{DM}T}$ is the reduced mass of the DM and the target. Therefore experiments with nuclear targets sensitive to recoil energies $\gtrsim$ keV, like [2–4], quickly lose sensitivity for DM masses lighter than about a GeV. In order to test sub-GeV masses, recent efforts have focused on different targets and/or detection strategies, see [6] for a review.

Recently, refs [7, 8] realised that another possibility to detect light DM is given by the higher-speed DM component upscattered by cosmic-rays (CRs). The existence of such a component is unavoidable as long as one assumes some DM interactions with the Standard Model (SM), which is nothing more than what is assumed in any direct detection experiment. The larger DM kinetic energy then implies much larger recoil energies at the Earth, so that unexplored regions of sub-GeV DM can be probed both in 'standard' experiments [7] like LUX, PandaX and XENON1T [2–4], and with completely different technologies like those available at large neutrino experiments [8].[1]

Focusing now on DM interacting mainly with nucleons, we improve over previous studies [7, 12–14] in two respects:

i) We derive new limits that are stronger than all existing ones [7, 12], by use of Super-K data of protons detected through their Cherenkov light [15, 16], and determine analogous sensitivities at Hyper-K and DUNE. The effectiveness of our strategy is due to the volumes and detected nucleon energies of these experiments, both larger than those of any other experiment previously used;

ii) We find regions of parameter space where our new limits and sensitivities are stronger not only than all other direct detection techniques, but also than all other constraints we know of (collider, cosmology, etc). We do so in two benchmark 'simplified' DM models, where we add a new particle to mediate the SM-DM interactions below the electroweak scale. Moreover we provide explicit completions of our simplified models above the electroweak scale, that are consistent with other constraints on the new model ingredients, and where our limits and sensitivities still win over all other existing ones. This is a first in the study of detection of CR-upscattered DM, and it allows to define a solid physics case to pursue this novel detection technique at large volume experiments, like neutrino detectors.

To appreciate the novelty of the second item, as well as to better introduce our work, it is useful to summarise here recent related papers. While refs. [7, 12] derived new limits on DM

---

[1]For earlier closely-related ideas see [9] for the effects of CR-DM scatterings on the CR spectrum, and [10, 11] for limits relying on the DM upscattered in the Sun.

interactions with nuclei, they only considered the case of constant cross section. As we stressed already in [8], it is important to include its energy dependence when comparing this detection technique with other ones, because they rely on different energy regimes. Refs [13,14] pursued this direction, but [13] did not compare the constraints with other ones, nor worried about finding an existence proof for their simplified models. Ref [14] did, but found that, in the specific model considered there, the parameter space probed by detection of CR-upscattered DM with Xenon-1T was already entirely excluded by other constraints.

The rest of this paper is organised as follows: in section 2 we present the simplified models of our interest and the resulting DM cross sections, in section 3 we derive new limits on light DM with Super-K data and new related sensitivities at Hyper-K and DUNE, in section 4 we discuss other constraints, in section 5 we prove that our new limits and sensitivities test theoretically-consistent parameter space. We conclude and discuss future directions in Section 6.

## 2 Dark Matter Interactions

We consider for definiteness DM as a Dirac fermion $\chi$, singlet under the SM gauge group, with mass $m_\chi$. We add to the Standard Model either a new scalar $\phi$ with mass $m_\phi$, or a new pseudoscalar $a$ with mass $m_a$, to act as mediators of DM-SM interactions via

$$\mathcal{L}_\phi = g_{\chi\phi}\, \bar{\chi}\chi\, \phi + g_{q\phi}\, \bar{q}q\, \phi \,, \tag{1}$$

$$\mathcal{L}_a = g_{\chi a}\, \bar{\chi} i\gamma_5 \chi\, a + g_{qa}\, \bar{q} i\gamma_5 q\, a \,, \tag{2}$$

where the index $q$ runs over the SM up, down and strange quarks. We foreshadow that CMB constraints either require DM to be asymmetric, or select the scalar model thanks to its $p$-wave suppressed DM annihilations, see Sec. 4.1. Now for our purpose of studying CR-upscattered DM, we have to convert the interaction with quarks of Eqs. (1) and (2), to that with nucleons and eventually with nuclei. In the following, we explain our procedure of the conversion for the scalar and pseudoscalar mediator cases separately.

Eqs. (1) and (2) give rise to different regimes of low-energy scattering of DM with nuclei $N$. At tree level, the leading non-relativistic operators in momentum space, induced by $\mathcal{L}_\phi$ and $\mathcal{L}_a$, are respectively (see e.g. [17])

$$\mathcal{L}_{N\phi} : \mathbb{1}\,, \qquad \mathcal{L}_{Na} : (\vec{s}_\chi \cdot \vec{q})(\vec{s}_N \cdot \vec{q})\,, \tag{3}$$

where $\vec{s}_N$ is the nucleus spin, $\vec{s}_\chi$ the DM one, $\vec{q}$ is the momentum exchanged in the $\chi - N$ scattering. Therefore 'standard' direct detection techniques test $\mathcal{L}_\phi$ via spin-independent searches, while they are insensitive to $\mathcal{L}_a$ at tree level. As we will see in Sec. 4, 'standard' DD techniques are sensitive to $\mathcal{L}_a$ at the one-loop level.

We anticipate that, instead, our detection technique is sensitive to both $\mathcal{L}_\phi$ and $\mathcal{L}_a$, because CRs have large kinetic energy and the scattering cross section does not suffer from suppression by velocity.

### 2.1 Scalar mediator

We write the matrix elements of the quark bilinear $\bar{q}q$, between nucleons $N = p, n$ with initial momentum $p_i$ and final momentum $p_f$, as

$$\langle N(p_f)|\bar{q}q|N(p_i)\rangle = \frac{m_N}{m_q} f_q^N\, G(-t)\, \bar{u}_N u_N \,, \tag{4}$$

where $u_N$ is the spinor wavefunction of the nucleon, $m_N$ and $m_q$ are the nucleon and quark masses, $t = (p_f - p_i)^2$ is the Mandelstam variable. Here $f_q^N$ is defined as

$$f_q^N = \frac{\langle N | m_q \bar{q} q | N \rangle}{m_N}, \tag{5}$$

at the leading order in the expansion for small $|t|$ after factorizing the spinor wavefunction, which we extract from [18] as

$$f_u^N = 1.99 \times 10^{-2}, \quad f_d^N = 4.31 \times 10^{-2}, \tag{6}$$

where we have averaged over proton and neutron. Note that the analysis of [18] is based on experimental data and is confirmed by [19, 20], while lattice QCD studies in general predict slightly smaller values of $f_u^N$ and $f_d^N$ (see, e.g., [21, 22]). The scalar nucleon form factor $G$ incorporates the finite size effect of nucleon, which reads [23]

$$G(q^2) = \frac{1}{(1 + q^2/\Lambda^2)^2}, \quad \Lambda = 770 \, \text{MeV}. \tag{7}$$

We then compute the differential scattering cross sections of DM with nuclei $A$. For the scalar mediator case, nucleons inside nuclei $A$ coherently contribute to the differential cross section. In our computation, we treat nuclei $A$ as a point particle when we evaluate the kinematical variables, and put the nuclei form factor to incorporate the finite size effect of nuclei $A$. The differential cross section is then given by[2]

$$\frac{d\sigma_\phi}{dK_f} = \frac{1}{K_{\max}} \frac{g_{\chi\phi}^2 g_{N\phi}^2}{16\pi s} \frac{\left(-t + 4m_\chi^2\right)\left(-t + 4m_A^2\right)}{\left(m_\phi^2 - t\right)^2} n_A^2 F_A^2(-t) \Theta\left(K_{\max} - K_f\right), \tag{8}$$

where $m_\chi$ is the DM mass, $K_f$ is the kinetic energy of the final state particle of interest (e.g. $f = \chi$ when we will compute the DM flux from CR upscatterings, $f = p$ when we will compute the proton recoil spectrum in detectors), $K_{\max}$ is its maximal value which will be computed case by case, $s$ is the usual Mandelstam variable, $n_A$ is the number of nucleons $N$ in the nucleus $A$ of interest (in this paper we will consider $n_A = 1, 4$ respectively for $A = {}^1\text{H}$ and $A = {}^4\text{He}$). As we mentioned, we evaluate $s$ and $t$ in this equation by the kinematics of nuclei $A$ as a whole. When $A$ is Hydrogen, $A = {}^1\text{H}$, this is simply a proton and hence the form factor is given by

$$F_{\text{H}}(q^2) = G(q^2). \tag{9}$$

When $A$ is Helium, $A = {}^4\text{He}$, the form factor is given by [23]

$$F_{\text{He}}(q^2) = \frac{1}{\left(1 + q^2/\Lambda_{\text{He}}^2\right)^2}, \quad \Lambda_{\text{He}} = 410 \, \text{MeV}. \tag{10}$$

The coupling $g_{N\phi}$ depends on how $\phi$ couples to quarks. In the following, we consider the isoscalar coupling $g_{u\phi} = g_{d\phi} = g_\phi$, and assume that couplings to the other quarks vanish. In this case, $g_{N\phi}$ is given as

$$g_{N\phi} = g_\phi \left( \frac{m_N}{m_u} f_u^N + \frac{m_N}{m_d} f_d^N \right). \tag{11}$$

---

[2]Another way of including the distribution of nucleons inside nuclei is discussed in [24], which also includes an incoherent contribution. We have checked that the differential cross section computed following [24] gives a similar constraint on the couplings. Since other uncertainties such as astrophysical ones have larger impacts on the final results, eq. (8) is enough for our purpose.

## 2.2 Pseudoscalar mediator

We write the matrix elements of the quark bilinear $i\bar{q}\gamma_5 q$, between nucleons $N = p, n$ with initial momentum $p_i$ and final momentum $p_f$, as

$$\langle N(p_f)|\bar{q}i\gamma_5 q|N(p_i)\rangle = \frac{m_N}{m_q} h_q\, G_a^q(-t)\, \bar{u}_N i\gamma_5 u_N\,, \tag{12}$$

where $h_q$ is defined as

$$h_q = \frac{\langle N|m_q \bar{q}i\gamma_5 q|N\rangle}{m_N}\,, \tag{13}$$

at the leading order in the expansion for small $|t|$ after factorizing the spinor wavefunction, that we extract from [25] as

$$h_{u-d} = 1.2, \quad h_{u+d} = 0.45\,, \tag{14}$$

for the isovector and isoscalar combinations, respectively. The coupling $h_s$ is also computed in [25], but we do not show it here since we take $g_{sa} = 0$ in our computation below.

The pseudoscalar nucleon form factor $G_a^q$ is evaluated by the partially conserved axial current (PCAC) relation as

$$G_a^q(-t) = G_A(-t) + \frac{t}{4m_N^2} G_P^q(-t) - 2\epsilon_q G_G(-t)\,. \tag{15}$$

We infer the axial vector form factor $G_A$ from the the isovector results in [25] as[3]

$$G_A(q^2) = \frac{1}{\left(1 + q^2/\Lambda_a^2\right)^2}, \quad \Lambda_a = 1.32\,\text{GeV}\,. \tag{16}$$

This value of $\Lambda_a$ is larger than the world average [26, 27], but is motivated by the recent MiniBooNE experiment [28] and the lattice results [25, 29]. As a check of the robustness of our results, we perform our analysis also for the world-averaged value $\Lambda_a = 1.03$ GeV. We anticipate that its impact is relatively mild, see Sec. 3.3 for more details. The induced pseudoscalar form factor $G_P^q$ is inferred from the pole dominance ansatz as

$$G_P^q(q^2) = G_A(q^2)\frac{C_q}{q^2 + M_q^2}\,, \tag{17}$$

where the parameters are extracted from [25] as

$$C_{u-d} = 4m_N^2, \quad M_{u-d} = m_\pi\,, \tag{18}$$

for the isovector combination, where $m_\pi$ is the pion mass, and

$$C_{u+d} = 0.90\,\text{GeV}^2, \quad M_{u+d} = 0.33\,\text{GeV}\,, \tag{19}$$

for the isoscalar combination. Finally $G_G$ denotes the anomaly form factor which originates from the $G\tilde{G}$-term in the PCAC relation. Since the isoscalar current is anomalous while the isovector current is not, $\epsilon_{u+d} = 1$ and $\epsilon_{u-d} = 0$. $G_G$ is computed, *e.g.*, in [30]. We have checked that $G_G$ is of little significance to our final result, and hence ignore it in the following for simplicity. Since all references we are aware of take the isospin symmetric limit, we assume

---

[3]The isoscalar result in [25] indicates an even larger value of $\Lambda_a$, but with a larger error. Hence we use the isovector result in our paper. If $\Lambda_a$ is indeed larger, our method becomes more sensitive to DM.

that the up and down quark masses are equal and are the arithmetic mean of their experimental values when we use the above formulas.

Since Helium does not have a net spin, we consider only the scattering with proton for the pseudoscalar mediator case in this paper. The differential scattering cross section with proton is given by

$$\frac{d\sigma_a}{dK_f} = \frac{1}{K_{\max}} \frac{g_{\chi a}^2 g_{Na}^2}{16\pi s} \frac{t^2}{\left(m_a^2 - t\right)^2} F_a^2(-t) \Theta\left(K_{\max} - K_f\right).$$ (20)

In the following, we consider the isoscalar coupling $g_{ua} = g_{da} = g_a$, and assume that couplings to the other quarks vanish. In this case, $g_{Na}$ and $F_a$ are given by

$$g_{Na} = g_a \frac{2m_N}{m_u + m_d} h_{u+d}, \quad F_a(q^2) = G_a^{u+d}(q^2),$$ (21)

where we assume the degenerate up and down quark masses as we mentioned above.

# 3 New limits and sensitivities

In this section we explain our method to compute the event rates at the terrestrial detectors, and show our new limits on the parameters of our simplified DM models.

## 3.1 DM upscattered by Cosmic Rays

Our galaxy is filled with high-energy CRs. Once these CRs have interactions with DM, they can upscatter DM such that DM acquires a large kinetic energy compared to its averaged value inside the galaxy. In the case of the hadrophilic DM, the CR protons and Heliums are the dominant source of such a high-energy component of the DM flux.

The DM flux $\Phi_\chi$ upscattered by the CRs is given by

$$\frac{d\Phi_\chi}{d\Omega} = \sum_A \int_{\text{l.o.s.}} dL \int dK_A \frac{d\Phi_A}{d\Omega} \frac{d\sigma}{dK_\chi} \frac{\rho_\chi}{m_\chi},$$ (22)

where $\Omega$ is the solid angle, "l.o.s." stands for the line-of-sight integral, $\Phi_A$ is the CR flux of the nuclei $A$ in the galaxy, $d\sigma/dK_f$ is the differential cross section between $A$ and $\chi$ which we explained in the previous section, and $\rho_\chi$ is the DM energy density in the galaxy. We include both $A = p$, He for the scalar mediator case, and include only $A = p$ for the pseudoscalar mediator case.[4]

In this paper, we assume for simplicity that the CR flux is homogeneous inside a leaky cylinder centered on the galactic center, and vanishes outside the cylinder. We can then simplify Eq. (22) as

$$\frac{d\Phi_\chi}{d\Omega} = \frac{J(b,l)}{m_\chi} \sum_A \int dK_A \frac{d\Phi_A}{d\Omega} \frac{d\sigma}{dK_\chi},$$ (23)

where $b$ and $l$ are the galactic latitude and longitude. The $J$-factor $J(b,l)$ contains the information of the DM- and CR-distributions within the galaxy, and is given by

$$J(b,l) = \int_{\text{l.o.s.}} dL \rho_\chi,$$ (24)

---

[4]In the pseudoscalar mediator case, although Helium does not have a net spin, DM may interact with individual nucleons if the energy transfer is large enough. Nevertheless we have checked that this contribution is negligible compared to the contribution from the CR protons.

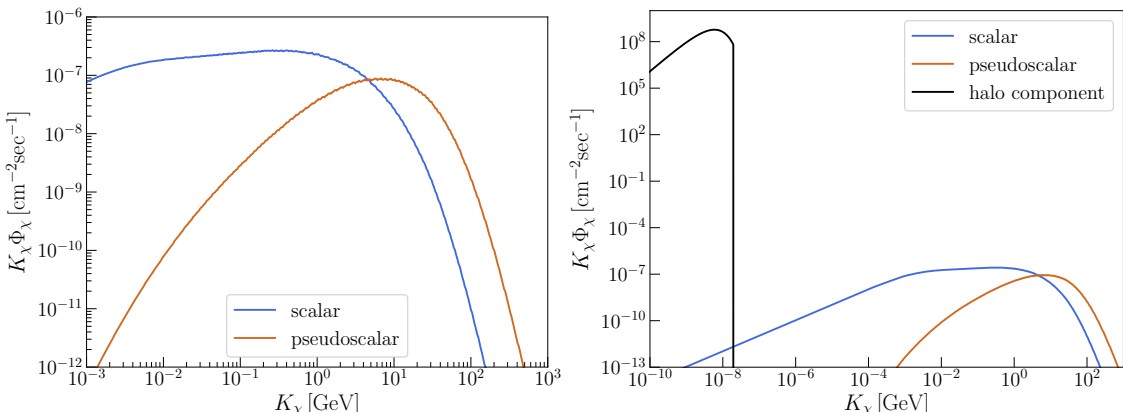

Figure 1: DM fluxes at the Earth for some benchmark values of the parameters. The right panel is a zoom-out of the left panel with the standard halo DM flux as a comparison. The DM and the mediator masses are fixed as 10 MeV and 1 GeV, respectively. The couplings are taken as $g_{\chi\phi}g_{u\phi} = g_{\chi\phi}g_{d\phi} = 0.1$ for the scalar mediator case, and $g_{\chi a}g_{ua} = g_{\chi a}g_{da} = 0.1$ for the pseudoscalar mediator case.

which depends on $b$ and $l$. In this formula, the integration is bounded by the leaky cylinder, whose radius $R$ and half width $h$ are fixed as $R = 10$ kpc and $h = 1$ kpc. We take the DM density profile as the NFW profile [31],

$$\rho_\chi(r) = \rho_\odot \frac{r_\odot (r_\odot + r_c)^2}{r (r + r_c)^2}, \tag{25}$$

where $r$ is the distance from the galactic center. We take $r_\odot = 8.5$ kpc, $r_c = 20$ kpc and, conservatively, $\rho_\chi(r = r_\odot) = 0.42$ GeV [32–34]. Note that our constraints only mildly depend on the choice of the DM density profile, since the $J$-factor is linear in $\rho_\chi$.

We input the CR flux as follows. For $K_A \leq 100$ MeV, we directly use Voyager's observation with interpolation [35]. The lower thresholds are taken as 22 MeV for proton and 10 MeV for Helium. We assume no CR flux below these energies simply because there is no observation. This assumption does not affect the DM event rates at the neutrino experiments, but may affect other DM direct detection experiments which have lower threshold recoil energy. For $100$ MeV $< K_A < 50$ GeV, we use the theoretical computation in [36]. Finally for $K_A > 50$ GeV, we use the fitting formula given in [37]. Note that the fitting formula in [37] underestimates the CR flux compared to Voyager's observation while [36] does not.

In Fig. 1, we show DM fluxes at the Earth's surface for some benchmark values of the parameters. We take $m_\chi = 10$ MeV, $m_\phi = 1$ GeV and $g_{\chi\phi}g_{u\phi} = g_{\chi\phi}g_{d\phi} = 0.1$ for the scalar mediator case, and $m_\chi = 10$ MeV, $m_a = 1$ GeV and $g_{\chi a}g_{ua} = g_{\chi a}g_{da} = 0.1$ for the pseudoscalar mediator case, respectively. As a comparison, we also show the standard halo DM flux in the right panel, which is given by

$$\Phi_\chi^{(\mathrm{halo})} = \frac{\rho_\odot}{m_\chi^2} c^2 f(v(K_\chi)), \tag{26}$$

where $c$ is the speed of light and $v(K_\chi) = \sqrt{K_\chi/2m_\chi}$. The velocity distribution function $f$ is given by

$$f(v) = \frac{1}{N} \sqrt{\frac{2}{\pi}} \frac{v^2}{\sigma^3} \exp\left(-\frac{v^2}{2\sigma^2}\right) \Theta(v_{\mathrm{esc}} - v),$$

$$N = \mathrm{Erf}\left(\frac{v_{\mathrm{esc}}}{\sqrt{2}\sigma}\right) - \sqrt{\frac{2}{\pi}} \frac{v_{\mathrm{esc}}}{\sigma} \exp\left(-\frac{v_{\mathrm{esc}}^2}{2\sigma^2}\right), \tag{27}$$

where $\Theta$ is the Heaviside theta function and Erf is the error function. We took $\sigma = 163$ km/sec (corresponding to the peak velocity 230 km/sec) and $v_{esc} = 600$ km/sec in Fig. 1. As one can see from the figure, although the overall normalization is much smaller compared to the standard halo DM flux, the CR-upscatterred DM flux is extended to far higher energy region. This feature makes it possible to probe this component of the DM by the neutrino experiments that have larger energy thresholds than the standard DM direct detection experiments.

## 3.2 Earth attenuation and events at detectors

The neutrino detectors are located deep inside the Earth in order to reduce backgrounds. As a result, DM can be scattered within the Earth before arriving at the neutrino detectors. We now explain how we implemented this Earth attenuation in our computation.

We approximate the energy loss at each scattering as its averaged value for simplicity. We also ignore the change of the direction of the DM velocity at each scattering. This treatment typically gives a slightly weaker bound on the DM cross section than a more refined treatment [38]. With these approximations, the DM kinetic energy $\bar{K}_\chi$ at a depth $z$ obeys the following differential equation:

$$\frac{d\bar{K}_\chi(z)}{dz} = -\sum_T n_T \int dK_T \, K_T \frac{d\sigma}{dK_T} \,, \tag{28}$$

where $n_T$ is the number density of a target particle $T$ inside the Earth. The initial condition is given by $\bar{K}_\chi(z=0) = K_\chi$ where the latter is the DM kinetic energy at the Earth's surface. Since the typical energy scale of the DM events at the neutrino experiments is higher than the nuclear energy scale, we ignore the nuclear structure of the target particles inside the Earth. Then the target particles are protons and neutrons, and we assume their ratio to be 1 : 1. The mass density is taken as $\rho_{p+n} = 2.7 \, \text{g/cm}^3$ [38], from which the number densities $n_p$ and $n_n$ are derived. We ignore the form factors of protons and neutrons, which results in a conservative constraint. The DM flux $\bar{\Phi}_\chi$ at a depth $z$ is then related to its value at the Earth's surface (computed by Eq. (22)) by

$$\bar{\Phi}_\chi(z) d\bar{K}_\chi(z) = \Phi_\chi dK_\chi \,, \tag{29}$$

which follows from the flux conservation.

As a comparison, we also show the constraints from Xenon1T for the scalar mediator case in the figures below. Since the Xenon1T detector is located deep inside the Earth, we also take into account the Earth attenuation in this case. Contrary to the neutrino experiment case, we treat the nuclei as a whole because of the lower threshold energy of Xenon1T. For simplicity, we assume that all the nuclei inside the Earth are with $A = 16$, corresponding to oxygen which is the dominant component. We have checked that the results do not change if we instead take $A = 12, 20$ or 24. We ignored the atomic form factor in this computation to be conservative.

Once we have implemented the Earth attenuation, we are ready to compute the DM event rates at the neutrino detectors. The direction-averaged proton event spectrum at a detector is given by

$$\frac{d^2 N_p}{dt \, dK_p} = N_T \int d\Omega \int dK_\chi \frac{d\Phi_\chi}{d\Omega} \frac{d\sigma}{dK_p} \left( \bar{K}_\chi(z), K_p \right) \,, \tag{30}$$

where $dN_p/dt$ is the number of the proton events per unit time, and $N_T$ is the total number of protons inside the detector. Note that the depth from the Earth's surface $z$ in this formula depends on the solid angle $\Omega$. The depth $z$ is the smallest when DM comes from right above the detector, while it is the largest when DM comes from the opposite side of the Earth. In the

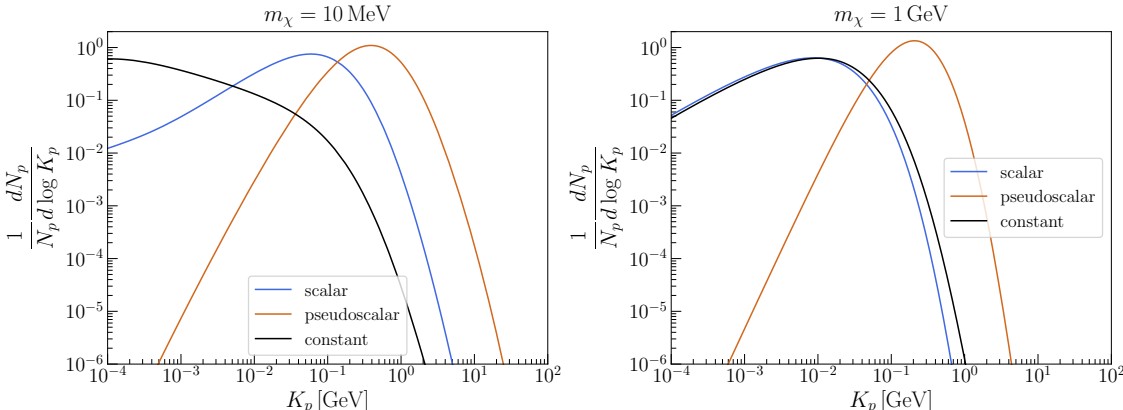

Figure 2: Normalized nucleon recoil event spectra as a function of the nucleon recoil energy without the Earth attenuation. The mediator mass is fixed as 1 GeV, and the DM mass is taken as 10 MeV and 1 GeV in the left and right panels, respectively. For comparison, the recoil event spectrum for the constant cross section case is also shown.

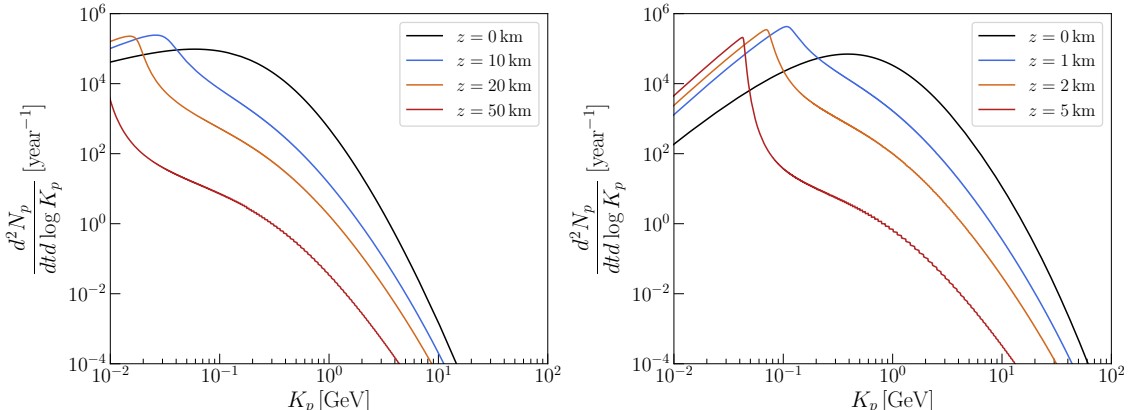

Figure 3: Dependence on depth of the detector. Left: the scalar mediator case with $m_\chi = 10\,\text{MeV}$, $m_\phi = 1\,\text{GeV}$, and $g_{\chi\phi}g_{u\phi} = g_{\chi\phi}g_{d\phi} = 0.1$. Right: the pseudoscalar mediator case with $m_\chi = 10\,\text{MeV}$, $m_a = 1\,\text{GeV}$, and $g_{\chi a}g_{ua} = g_{\chi a}g_{da} = 0.1$. Here we take the number of the target particle as that of Super-K, *i.e.*, $N_T = 7.5 \times 10^{33}$.

case of DUNE, we can also use neutrons as our signal, and hence we use the same formula but replacing $p \rightarrow n$ to compute the neutron event spectrum.

In Fig. 2, we show the normalized recoil spectra $(1/N_p)dN_p/d\log K_p$ without the Earth attenuation. In this figure, we fix the DM and mediator masses as 10 MeV and 1 GeV, respectively. For comparison, the recoil event spectrum for the constant cross section case considered in [7, 12] is also shown. As one can see, the spectra for the scalar and pseudoscalar mediator cases are extended to high energy region compared to the constant cross section case, especially when $m_\chi$ is smaller. It is because the cross section grows with energy for the scalar and pseudoscalar cases, as long as the energy transfer is smaller than the mass of the mediator. Because of this feature, an experiment with a larger energy threshold such as the neutrino experiments tend to be more sensitive to the cases with the massive mediators.

**Comments on the Earth attenuation.** In order to visualise the Earth attenuation effects, we plot in Fig. 3 the spectra per unit time $d^2N_p/dt\,dK_p$ at different values of the depth $z$. The left panel is the scalar mediator case with $m_\chi = 10\,\text{GeV}$, $m_\phi = 1\,\text{GeV}$, and $g_{\chi\phi}g_{u\phi} = g_{\chi\phi}g_{d\phi} = 0.1$, and the right panel is the pseudoscalar mediator case with

$m_\chi = 10\,\text{MeV}$, $m_a = 1\,\text{GeV}$, and $g_{\chi a}g_{ua} = g_{\chi a}g_{da} = 0.1$, respectively. It is clear that the Earth attenuation depletes the DM kinetic energy and hence the peak of the recoil spectrum is shifted toward lower energies as $z$ becomes larger. If $z$, or equivalently the coupling,[5] is too large, the Earth attenuation is so effective that no DM flux reaches the detector. As a result, our constraints on the coupling are bounded both from below and from above.

The left- and right-hand plots in Fig. 3 also display a qualitative difference between the scalar and pseudoscalar mediator cases, namely that the peak shifted to lower energies is more pronounced in the latter case than in the former. This feature is due to the fact that the pseudoscalar-mediator cross section, cf. Eq. (20), goes to zero in the limit of no exchanged energy, so that below a certain $K_p$ Earth attenuation becomes irrelevant and the DM flux accumulates there. The scalar-mediator cross section, cf. Eq. (8), instead goes to a constant in the limit of no exchanged energy, so that the peak feature at low $K_p$ is less pronounced. We have explicitly checked that, as expected from our understanding, the peak feature actually disappears in the scalar-mediator case for a value $m_\chi = 3\,\text{GeV}$ (thus different than those in Fig. 3), for which the constant limit in the cross section kicks in for larger values of the exchanged energy.

Another very interesting point, concerning Earth attenuation, is that it can make some constraints stronger, rather than weaker. Let us imagine the situation that a detector is sensitive to the energy region $K_\text{min} < K_p < K_\text{max}$, which satisfies $K_\text{max} < K_\text{peak}$, where $K_\text{peak}$ is the peak energy of the recoil spectrum without the Earth attenuation. In this case, the Earth attenuation can help to probe the DM by shifting the peak energy to lower such that it falls into the range $K_\text{min} < K_\text{peak} < K_\text{max}$, before it completely depletes the DM flux. Since the peak energy is $\mathcal{O}(100\,\text{MeV–1\,GeV})$ without the Earth attenuation, this effect makes the constraints stronger for experiments with lower threshold energy such standard DD experiments. On the other hand, this effect does not strengthen the constraints of of experiments with larger threshold energies like Super-K and Hyper-K.

## 3.3 New limits and sensitivities

We now derive our new limits on the DM models by Super-K, and also future sensitivities of Hyper-K and DUNE. We derive the limits and sensitivities in the following way. In deriving them, we use the DM flux averaged over the solid angle.

Super-K measured $N_d = 16$ downgoing and $N_u = 13$ upgoing events for 2287.3 days in the energy range $0.485 < K_p/\text{GeV} < 3.17$ (corresponding to the momentum range $1.07 < p_p/\text{GeV} < 4$) [16]. Since our signal is expected to be downgoing in the range of the cross section of our interest, we derive our limits by requiring that

$$\epsilon_\text{SK} N_p^\text{SK}\left(0.485 < K_p/\text{GeV} < 3.17\right) < (N_d - N_u) + 2\sqrt{N_d} = 11\,, \tag{31}$$

where $\epsilon_\text{SK} = 0.55$ is the efficiency [16]. In our computation, we assume that all protons inside the detector contribute to the events because of the high energy threshold, and hence take $N_T^\text{SK} = 7.5 \times 10^{33}$, corresponding to an effective volume of 22.5 ktons. The depth of the Super-K detector from the Earth's surface is $z_\text{SK} = 1\,\text{km}$.

Hyper-K is supposed to start taking data in 2027 with its fiducial volume 187 ktons [39]. In order to estimate the upper limit, we assume 5 years of data taking with the same number of background event rate as Super-K. Therefore, we require that

$$\epsilon_\text{HK} N_p^\text{HK}\left(0.485 < K_p/\text{GeV} < 3.17\right) < 2\sqrt{\frac{N_u + N_d}{2}\frac{5\,\text{yrs} \times 187\,\text{kt}}{6.3\,\text{yrs} \times 22.5\,\text{kt}}} = 19.6\,, \tag{32}$$

---

[5] Note that $z$ appears only in the combinations $g_{\chi\phi}^2 g_{u\phi}^2 z$ or $g_{\chi a}^2 g_{ua}^2 z$.

where we assume the same signal energy range as Super-K and take $\epsilon_{\text{HK}} = 0.55$. We take the number of the target particle as $N_T^{\text{HK}} = 6.2 \times 10^{34}$ and the depth as $z_{\text{HK}} = 0.65\,\text{km}$.

The DUNE detector will consist of 4 modules, each with 10 ktons of fiducial volume. The first two are expected to be completed by 2024, and be operational by 2026 [40]. Assuming the other two will be installed and operational by 2027, we show sensitivities for 5 years of data taking. Since DUNE relies on the scintillation light, not the Cherenkov light, the threshold energy can be lower than Super-K and Hyper-K. We take it as $K_p > 50\,\text{MeV}$ [41]. We take the upper energy threshold as 10 GeV, although the result is insensitive to this choice as long as it is much larger than 50 MeV. In this energy range, DM dominantly scatters with individual nucleons [42], and hence we take the number of the target particles as $N_T^{\text{DN}} = 2.4 \times 10^{34}$ where we summed all protons and neutrons. Concerning background events, we are not aware of any detailed study that is straightforwardly applicable to our case. Therefore, we simply draw the lines by requiring

$$N_{p+n}^{\text{DN}} \left( 0.05 < K_p/\text{GeV} < 10 \right) = 30\,, \tag{33}$$

in the figures below, where we assume $\epsilon_{\text{DN}} = 1$, and leave a detailed study on background events as a future work. Finally, the depth of the detector is taken as $z_{\text{DN}} = 1.5\,\text{km}$ [40].

Our results are displayed in Figure 4. There we also show limits on CR-upscattered DM from XENON1T data, computed following [7] [6], as well as other experimental limits on the models, discussed below in Sec. 4. Their comparison with our new constraints and sensitivities shows that our method can probe a parameter region which is not yet probed by any other experiments. We also show limits from KamLAND data [43] and sensitivities of JUNO [44], computed following [12]. KamLAND observed one event in the bin [13.5 MeVee, 20 MeVee] within 123 kton-day where MeVee is an abbreviation for MeV electron equivalent, and hence we require the DM events smaller than three in this bin. We assume that JUNO has the same background event rate as KamLAND and compute the sensitivity of the bin [13.5 MeVee, 20 MeVee] for the exposure of 5 years with a 20 kton detector. Finally we also compute limits from Borexino [45]. Following [7], we use $N_p = 3.2 \times 10^{31}$, an exposure of 1.282 years and then impose that the number of events be smaller than 2.44 in the range 21.6 MeV $< K_p <$ 24.9 MeV. This range follows from the interval [12.5 MeVee, 15 MeVee], where we have inferred the upper limit from [45] with the detector information also taken from [46]. The resulting limits are always slightly weaker than the ones of KamLAND, thus to avoid clutter we refrain from displaying them in Figure 4. Of course, limits from XENON1T, KamLAND and Borexino and sensitivities from JUNO have been derived with the same astrophysical inputs used in the rest of this work.

**Discussion.** The limits and sensitivities that rely on CR-upscattered DM extend to DM masses much smaller than those shown in 4, we do not display that region just because it is anyway excluded by BBN. They also suffer some astrophysical uncertainties, like the precise value of the local DM density and the region of the Milky Way where our input CR spectra are valid. The impact of these uncertainties on limits and sensitivities on CR-upscattered DM, however, is limited: the signal events scale like the square of the cross section (because one hit is necessary to upscatter the DM, and one to detect it), and so as the fourth power of the coupling products on the vertical axis of Fig. 4. The signal events also scale linearly in the astrophysical quantities mentioned above, so that an error as large as a factor of 2 in, e.g., the DM local density, would translate into a shift of less than 20% in the contours shown in Fig. 4.

---

[6]As a check, we have computed the XENON1T constraint with the same inputs ($J$-factor etc) of [14] and we reproduced their limits. We have done the same for the constant cross section case, with the same inputs of [7], and we obtained a lower bound on the cross section a factor of $\sim \sqrt{2}$ weaker than [7]. We could not identify the origin of this difference.

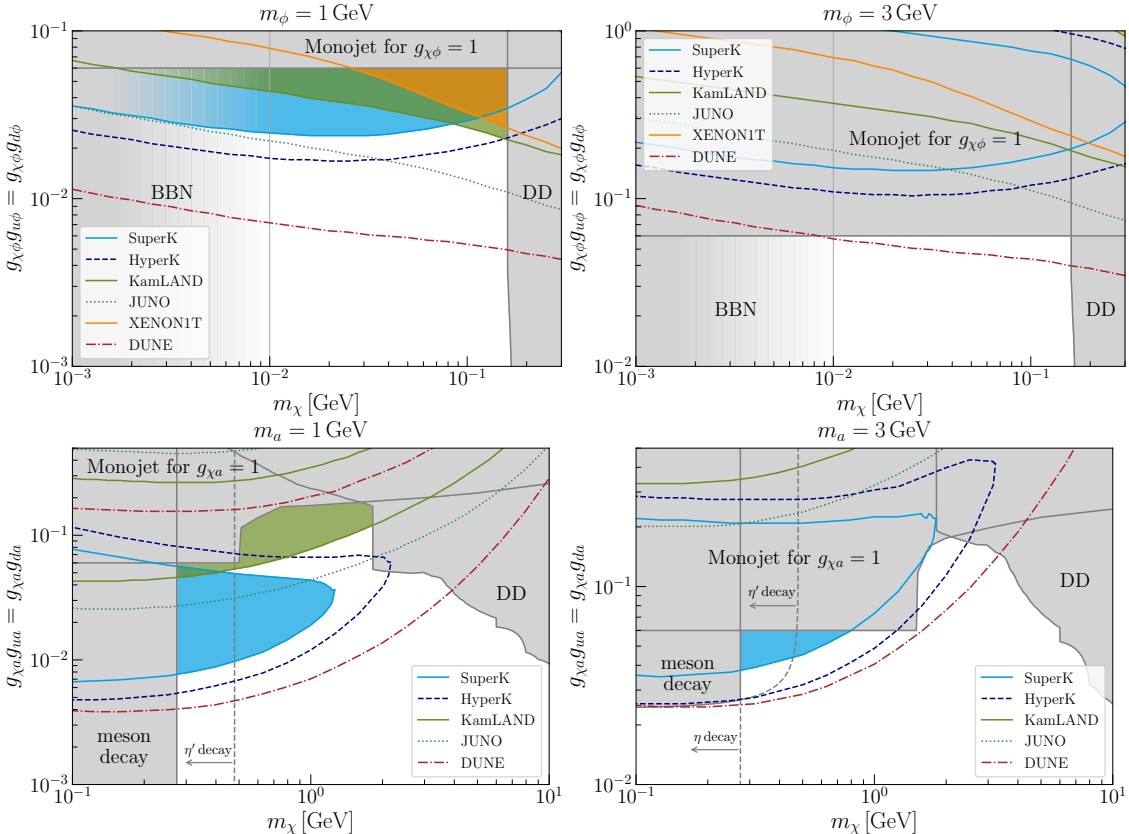

Figure 4: Limits and sensitivities in the plane of couplings vs DM mass, for mediator masses of 1 GeV(left) and 3 GeV(right). Coloured limits and sensitivities are those that rely on the DM subcomponent that is inevitably up-scattered by cosmic rays, among those the new limits and sensitivities proposed in this work are Super-K (blue), Hyper-K (dark blue dashed) and DUNE (red dot-dashed). The use of XENON1T data have been proposed in [7], the use of KamLAND and future JUNO data in [12]. We have recast them on the models considered here, with a procedure consistent with that used for our new results, see Sec. 3. Shaded gray areas are, from left to right, limits from BBN, our recast of monojet searches, 'standard' DD, see Sec. 4. The BBN limits are shaded differently than the other ones because, for 1 MeV $\lesssim m_\chi \lesssim$ 10 MeV, one could alleviate them by switching on a coupling to neutrinos, in a way that does not affect any of the other phenomenology presented here, see text for more details. Upper: scalar mediator. Lower: pseudoscalar mediator. In the latter case, limits from meson decay depend further on the coupling to the strange quark. Since this negligibly affects the rest of the phenomenology, we shade only the area that is excluded independently of its value, and display as a dashed lines the ones that are excluded for $g_{sa} = g_{ua}$ ($\eta'$) and $g_{sa} = -2g_{ua}$ ($\eta$).

Coming now to comments specific to our new limits and sensitivities, one sees that in the pseudoscalar case DUNE is expected to give marginal improvements with respect to Super-K and Hyper-K, for sub-GeV DM masses. Instead, in the scalar case and over the entire $m_\chi$ range that we plot, DUNE is expected to improve substantially over Super-K and Hyper-K. This can be explained by noticing that the event recoil spectrum, in the scalar case, stays sizable at lower energies, so that the lower threshold energy of DUNE constitutes an advantage. In the pseudoscalar case instead the recoil spectrum drops significantly at lower energies (see e.g. Fig. 2), explaining why one does not see an analogous improvement at DUNE. Limits from KamLAND and sensitivities from JUNO can be understood in the same way. This property constitutes a concrete example of the necessity to use explicit models, with the resulting relativistic and momentum-dependent cross sections, when comparing the reach of different

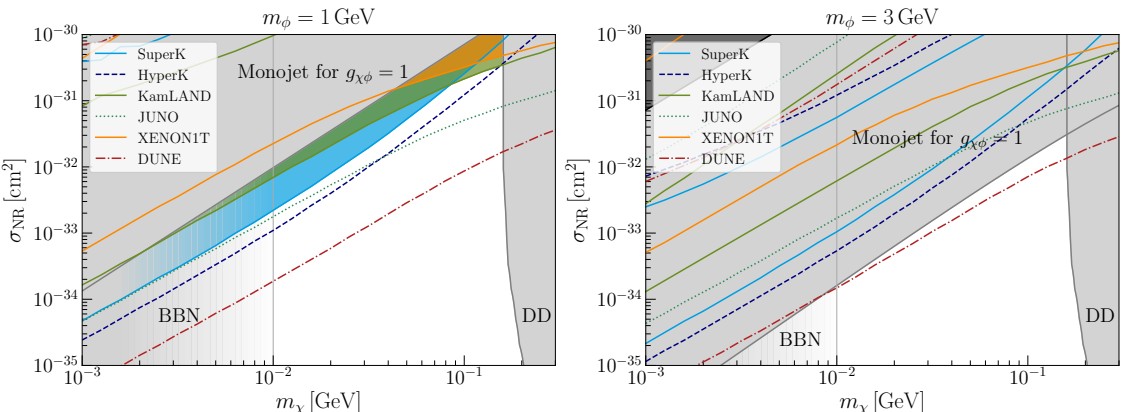

Figure 5: Limits and sensitivities on the scalar case in the plane of non-relativistic cross section vs DM mass for two different values of the scalar mediator mass. Lines and shadings as in Fig. 4. The dark gray region at the upper left corner for $m_\phi = 3\,\text{GeV}$ is with $g_{\chi\phi} g_{u\phi} \geq 4\pi$.

experiments to CR-upscattered DM. The lower energy threshold of DUNE also explains why its sensitivity improves a lot for DM masses above a GeV, with respect to Super-K and Hyper-K. We have checked that, in the pseudoscalar case using an axial mass $\Lambda_a = 1.03\,\text{GeV}$ instead of $\Lambda_a = 1.32\,\text{GeV}$ (see Eq. (16) and the related discussion) worsens our limits on $g_{\chi a} g_{ua}$ by a factor of order 10%.

Concerning finally XENON1T, it is insensitive to the pseudoscalar mediator case because of its low detection energy. The cross section grows with energy for the massive mediator cases, hence as visible in Fig. 4 neutrino experiments are more sensitive to CR-upscattered DM than standard direct detection experiments, because they have larger detection energies.

## 3.4 Limits and sensitivities on cross sections

In Figure 5 we translate our results, for the scalar mediator case, to limits and sensitivities on the spin-independent non-relativistic cross section for DM scattering with protons

$$\sigma_{\text{NR}} = C \frac{g_{\chi\phi}^2 g_{u\phi}^2}{\pi} \frac{\mu_{\chi p}^2}{m_\phi^4}, \qquad C = \left( \frac{m_p}{m_u} f_u^p + \frac{m_p}{m_d} f_d^p \right)^2 \simeq 300, \qquad (34)$$

where $\mu_{\chi p} = m_\chi m_p / (m_\chi + m_p)$, see Sec. 4.3 for more details. First, this is a quantity that could be more familiar to some readers. Second, this now allows us to comment on why the use of simplified models is, for our new constraints, more appropriate than the one of an EFT. Our new strategy probes parameter space allowed by other limits, like monojet at the LHC and standard DD, only for mediator masses around a GeV, while only DUNE will probe new parameter space for mediator masses of 3 GeV. When the mediator mass is around a GeV, the energy dependence in the denominators of the cross sections (see e.g. Eq. (8)) cannot be neglected as it is in an EFT approach. Indeed, if it could be neglected, then our new limits in the left- and right-hand plots of Fig. 5 would coincide, while they are different by a factor as large as $\sim 2$ in the case of Super-K and Hyper-K. They are still different, but slightly less, in the case of DUNE, as can be understood by the fact that the energies relevant for DUNE are smaller than those relevant for Super-K and Hyper-K. The even lower energies relevant for KamLAND, JUNO and especially XENON1T are, then, the reason why those lines almost does not change by changing the mediator mass from 1 to 3 GeV. To summarise, our new limits and sensitivities really test new parameter space in a region where the use of an EFT is not fully justified, so that they should be accompanied by the value of the mass of the mediator of the interactions between DM and the SM.

We conclude this section with a comment on the use of our limits and sensitivities, and more in general on all those coming from CR-upscattered DM. For detection techniques that involve only energies much lower than the DM and nucleon masses, one typically finds several models that map to a given assumption on the energy-dependence of the cross section. For example, in 'standard' DD, the model we studied of fermion DM with scalar mediators realises a cross section which is constant in the energies relevant for those experiments. Explicitly, at those experiments the Mandelstam variable $t$ is negligible with respect to $m_\chi$, $m_A$ and $m_\phi$ in Eq. (8). This is not the case for CR-upscattered DM, so that the cross-section depends on the relevant energies. Therefore, the use of the latter limits and sensitivities, shown in colour in Figure 5, is perfectly consistent with the assumption of cross section constant at smaller energies. More than that, our results proves that they are actually the stronger existing constraint over a significant range of DM masses. On the contrary, the assumption of a cross section constant in the energies relevant for CR-upscattered DM, as done in a relatively wide portion of the literature on the subject, at present does not stand on the same solid footing. While that regime in energy could be approximately realised in the case of scalar DM with scalar mediated interactions, one should assess if the resulting limits on CR-upscattered DM (see e.g. [12]) test parameter space that is allowed by monojet searches, whose shape in the $\sigma_{\mathrm{NR}} - m_\chi$ plane would be the same as in Fig. 5.

# 4 Other limits

We describe here other constraints on DM with interactions with nuclei, and constraints specific to the models of Eqs. (1) and (2). They are shown in Figures 4 and 5.

## 4.1 Cosmology

If the dark sector is light, of order MeV, it can modify the thermal history in the early universe. Recent studies of the BBN and CMB constraints on MeV-scale dark matter thermally coupled to the SM plasma in the early universe include [47–55].

**BBN.** In our benchmark points, we assume that $m_\phi, m_a \sim \mathcal{O}(1)\,\mathrm{GeV}$, and that these mediators decay well before BBN, thus not affecting BBN nor CMB. However, the dark matter can affect BBN (and CMB, see later) and hence can be constrained if $m_\chi \lesssim O(10)\,\mathrm{MeV}$. Although the precise bound depends on the specific couplings to the SM particles [55], to be conservative we simply assume that BBN requires $m_\chi > 10$ MeV. By switching on a coupling of the mediators to neutrinos, with appropriate size, one could relax the BBN limit to $m_\chi \gtrsim O(1)\,\mathrm{MeV}$ [53,55], and this is why we extend the range of some Figures to $m_\chi = 1$ MeV and we shade BBN differently than other constraints.

**Dark Matter abundance.** By using the only couplings of Eqs. (1), (2) and for the coupling values of our interest, DM is in chemical equilibrium with the early-universe bath as long as it contains quarks. After the QCD phase transition takes place, DM pairs could still annihilate into two (scalar-mediator) or three (pseudoscalar-mediator) pions. Therefore, achieving the correct relic abundance via thermal freeze-out would require $m_\chi \gtrsim m_\pi$ or $m_\chi \gtrsim 3m_\pi/2$. One would then conclude that smaller masses overclose the universe, and are therefore excluded. We note however that a small coupling of the mediator to neutrinos (perhaps even to electrons in the scalar case) would allow to achieve the correct relic abundance via thermal freeze-out even for DM masses smaller than $O(m_\pi)$, without worsening the BBN constraints discussed so far, and without introducing other ones. We then refrain from displaying lines corresponding

to the correct relic abundance in Figs. 4 and 5, because they would depend on free parameters unrelated to the phenomenology presented there. In addition, if the DM annihilation cross section is larger than the thermal one (e.g. for $m_\chi$ enough larger than $m_\pi$ or if we add a small coupling to neutrinos), then the symmetric DM component annihilates away leaving only a possible initial asymmetric component, dependent on other more UV physics.

**CMB.** Since CMB constraints are absent when DM is asymmetric enough [56,57], we refrain from displaying them in Figs. 4 and 5, because as we have seen they can be dealt with by physics that is unrelated to the phenomenology that is the focus of this work. If one was instead interested in the case where the DM abundance is set by the only couplings of Eqs. (1) and (2), so that as we have argued $m_\chi > O(m_\pi)$, then

i) in the scalar mediator case, CMB would not exclude the currently allowed regions of parameter space, because the DM annihilations into mesons and quarks (and mediators, if the masses allow) are all *p*-wave suppressed (see e.g. [58]);

ii) in the pseudoscalar mediator case, CMB would exclude the model, because DM annihilations are instead in *s*-wave. Exceptions exist, e.g. if the thermal abundance is set by resonant annihilations [59], but we do not explore them further here.

## 4.2 Collider

**Monojet at LHC.** Dark Matter pairs can be produced at the LHC together with SM radiation, in processes like $q\bar{q} \to g\phi(a) \to g\chi\bar{\chi}$ where, depending on its mass, the mediator $\phi$ (*a*) can either be off-shell, or be produced on-shell and then decay to DM. Thus the parameter space can be constrained by searches of monojet plus missing energy like [60, 61], that we recast here to our model.

We use FeynRules 2.3.32 [62] to generate the model files, and MadGraph 5 2.7.3 [63, 64] to generate events $pp \to \chi\bar{\chi}$+jets, up to three jets. We then shower with Pythia 8.2 [65] switching on MLM merging [66], and use Delphes 3.4.1 [67] to simulate the ATLAS detector. We impose weak cuts at generation level, the most important ones being htjmin = 180 and transverse momentum of each DM particle larger than 60 GeV. We then use Madanalysis5 [68] to reject events after detector simulation that do not satisfy the cuts described in [60]: we demand that the leading jet transverse momentum, $p_T(j_1)$, be larger than 150 GeV, that there is not a fifth jet with $p_T(j_5)$ larger than 30 GeV, and that the missing transverse energy (MET) be larger than 200 GeV. This defines the inclusive region called IM0 in [60] (we find that additional cuts on the azimuthal separation between the leading jet and MET have no further impact in reducing our events). Regions from IM1 to IM12 are then defined by the cut MET/GeV > 250, 300, 350, 400, 500, 600,... 1200. By using the merged cross sections given by Pythia[7] and the efficiencies of the various cuts given by MadAnalysis 5, we finally obtain the signal cross sections after cuts for each of the 13 regions above.

We then compare these cross sections with the model-independent limits, on the visible (i.e. multiplied by acceptance and efficiency) cross sections, given in Table 7 of [60] for each of the regions IM0,...IM12 above. We translate the strongest of these limits into an upper limit on $g_{u\phi} = g_{d\phi}$ and $g_{ua} = g_{da}$. By doing so we are implicitly assuming that our simulation accounts for the entire acceptance and efficiency: in this sense our limits can be seen as aggressive ones, which is a conservative choice from the point of view of determining the parameter space open

---

[7]The output files tag_1_merged_xsecs.txt contain three merged cross sections, one for each of the merging scales of 45, 67.5 and 90 GeV. For definiteness we use the cross section corresponding to 67.5 GeV, which very often lies in between the cross sections corresponding to 45 and 90 GeV. The three cross sections differ by at most 20%, which translates to an uncertainty of $\sim$ 10% on our limits on the couplings.

for the new searches at neutrino experiments proposed in this paper. We repeat the procedure for mediator masses of 1 and 3 GeV, and DM masses from 0.01 to 10 GeV, finding the limits displayed in Figs. 4 and 5, which for $m_\chi < m_{a,\phi}/2$ read $g_{ua,u\phi} = g_{da,d\phi} \lesssim 0.06$. In the pseudoscalar case and for a selected subset of parameters, we repeat the procedure also for $g_{u\phi} = -g_{d\phi}$ and $g_{ua} = -g_{da}$, and find comparable limits. This was expected because that relative sign matters only in interference diagrams, which are present only for multiple jet emissions and thus are less important. It was also expected that our limits are the same in the scalar and pseudoscalar case and that, for $m_\chi < m_{a,\phi}/2$, they are independent of the DM mass, because DM is produced from mediator decays and so what matters is the mass of the latter. As a check of our results, we found comparable limits also when we generated events with stronger generation level cuts (`htjmin = 380` and transverse momentum of each DM particle larger than 120 GeV), so to have more statistics in the signal regions with more aggressive cuts. Finally, we checked our limits with another simulation carried out independently.

**Meson invisible decay.**   The hadrophilic mediators and dark matter particles can be produced, and hence can be probed, by meson decays. Since the mediators couple to light quarks, light mesons such as $\pi$, $K$ and $\eta$ are relevant in our case. We are interested in the relatively heavy mediator mass, $m_\phi, m_a \gtrsim 1\,\mathrm{GeV}$, and hence these mediators cannot be directly produced from the decay of $\pi$, $K$ and $\eta$. Still the mesons can decay into dark matter particles and one can constrain the parameter region from their invisible decay. This is particularly relevant for the pseudoscalar mediator case since the pseudoscalar mediator has the same quantum numbers as neutral mesons and can mix with them.

Because of the coupling with light quarks (2), the pseudoscalar mediator $a$ can mix with the neutral mesons. The chiral Lagrangian with the mediator $a$ is given by

$$\mathcal{L} = \frac{f_\pi^2}{4}\mathrm{tr}[\partial_\mu U \partial^\mu U^\dagger] + \frac{f_\pi^2 m_\pi^2}{4\bar{m}}\mathrm{tr}[M^\dagger U + U^\dagger M] - \frac{1}{2}f_\pi^2 m_{\mathrm{anomaly}}^2(\mathrm{Im}\log\det U)^2, \tag{35}$$

$$U = \exp\left[-\frac{\sqrt{2}i}{f_\pi}\begin{pmatrix} \pi^0/\sqrt{2} + \eta/\sqrt{6} + \eta'/\sqrt{3} & \pi^+ & K^+ \\ \pi^- & -\pi^0/\sqrt{2} + \eta/\sqrt{6} + \eta'/\sqrt{3} & K^0 \\ K^- & \bar{K}^0 & -2\eta/\sqrt{6} + \eta'/\sqrt{3} \end{pmatrix}\right], \tag{36}$$

$$M = \mathrm{diag}(m_u + ig_{ua}a, m_d + ig_{da}a, m_s + ig_{sa}a). \tag{37}$$

Here $f_\pi = 93$ MeV, $\bar{m} = (m_u + m_d)/2 = 3.5$ MeV [69] and the eta prime decay constant is taken to be equal to the pion decay constant. The last term in Eq. (35) gives an anomalous mass term to $\eta'$. By expanding this Lagrangian, one finds the following mass mixing terms between $a$ and the mesons

$$\mathcal{L}_{\mathrm{mix}} = -\frac{f_\pi m_\pi^2}{\bar{m}}\left[\frac{1}{2}(g_{ua} - g_{da})\pi^0 a + \frac{1}{2\sqrt{3}}(g_{ua} + g_{da} - 2g_{sa})\eta a + \frac{1}{\sqrt{6}}(g_{ua} + g_{da} + g_{sa})\eta' a\right]. \tag{38}$$

In the limit of small $g$'s, the mixing angles are given as

$$\theta_{\pi a} \simeq \frac{1}{m_a^2 - m_\pi^2}\frac{f_\pi m_\pi^2}{2\bar{m}}(g_{ua} - g_{da}), \tag{39}$$

$$\theta_{\eta a} \simeq \frac{1}{m_a^2 - m_\eta^2}\frac{f_\pi m_\pi^2}{2\sqrt{3}\,\bar{m}}(g_{ua} + g_{da} - 2g_{sa}), \tag{40}$$

$$\theta_{\eta' a} \simeq \frac{1}{m_a^2 - m_{\eta'}^2}\frac{f_\pi m_\pi^2}{\sqrt{6}\,\bar{m}}(g_{ua} + g_{da} + g_{sa}). \tag{41}$$

The total decay widths of neutral mesons are $\Gamma_{\pi_0} = 7.7 \times 10^{-3}\,\text{keV}$ ($\tau_{\pi^0} = 8.5 \times 10^{-17}\,\text{s}$), $\Gamma_\eta = 1.3\,\text{keV}$ and $\Gamma_{\eta'} = 1.9 \times 10^2\,\text{keV}$ [69]. The constraints on their invisible decay are $\text{Br}(\pi^0 \rightarrow \text{invisible}) < 4.4 \times 10^{-9}$ [70], $\text{Br}(\eta \rightarrow \text{invisible}) < 1.0 \times 10^{-4}$ and $\text{Br}(\eta' \rightarrow \text{invisible}) < 6 \times 10^{-4}$ [69]. The invisible decay rates are given by

$$\Gamma_{\pi^0 \rightarrow 2\chi} = \frac{g_{\chi a}^2 \theta_{\pi a}^2 m_{\pi^0}}{8\pi} \sqrt{1 - \frac{4m_\chi^2}{m_{\pi^0}^2}}, \tag{42}$$

$$\Gamma_{\eta \rightarrow 2\chi} = \frac{g_{\chi a}^2 \theta_{\eta a}^2 m_\eta}{8\pi} \sqrt{1 - \frac{4m_\chi^2}{m_\eta^2}}, \tag{43}$$

$$\Gamma_{\eta' \rightarrow 2\chi} = \frac{g_{\chi a}^2 \theta_{\eta' a}^2 m_{\eta'}}{8\pi} \sqrt{1 - \frac{4m_\chi^2}{m_{\eta'}^2}}. \tag{44}$$

We then obtain the following constraints on the couplings:

$$|g_{ua} - g_{da}|g_{\chi a} < 3 \times 10^{-7} \times \frac{|m_a^2 - m_{\pi^0}^2|}{1\,\text{GeV}^2} \left(1 - \frac{4m_\chi^2}{m_{\pi^0}^2}\right)^{-1/4}, \tag{45}$$

$$|g_{ua} + g_{da} - 2g_{sa}|g_{\chi a} < 6 \times 10^{-4} \times \frac{|m_a^2 - m_\eta^2|}{1\,\text{GeV}^2} \left(1 - \frac{4m_\chi^2}{m_\eta^2}\right)^{-1/4}, \tag{46}$$

$$|g_{ua} + g_{da} + g_{sa}|g_{\chi a} < 9 \times 10^{-3} \times \frac{|m_a^2 - m_{\eta'}^2|}{1\,\text{GeV}^2} \left(1 - \frac{4m_\chi^2}{m_{\eta'}^2}\right)^{-1/4}. \tag{47}$$

Thus, the meson invisible decays in general put a severe constraint on the pseudoscalar mediator model. However, we note that some of these constraints can be evaded once we switch on the couplings to the strange quark. For instance, one can consider the SU(3) singlet case, $g_{ua} = g_{da} = g_{sa}$. In this case, one can evade the mixing with $\pi^0$ and $\eta$, and only the $\eta'$ invisible decay applies. Alternatively one can take $g_{sa} = -2g_{ua} = -2g_{da}$, which makes only the $\eta$ invisible decay relevant. Since our constraint is insensitive to $g_{sa}$, we shade the parameter region covered by both the $\eta$ and $\eta'$ invisible decays in Figure. 4. The individual constraints are indicated as the dashed lines, implying that it depends on $g_{sa}$ which line is actually relevant. For these lines, we assume that $g_{sa} = g_{ua} = g_{da}$ for the $\eta'$ invisible decay and $g_{sa} = -2g_{ua} = -2g_{da}$ for the $\eta$ invisible decay, respectively.

Yet another option would be the isovector coupling $g_{ua} = -g_{da}$ and $g_{sa} = 0$. In this case, the $\pi^0$ invisible decay puts a severe constraint for $m_\chi < m_{\pi^0}/2$, but there is no meson decay constraints for $m_\chi > m_{\pi^0}/2$. However, our limits and sensitivities at neutrino experiments also get weaker due to the exact cancellation of the form factor in the large $|t|$ limit (see Eqs. (17) and (18)). Since we could not derive a meaningful constraint on the couplings within our treatment, we do not seek this possibility any further.[8]

If the mediators are lighter, $m_\phi, m_a \lesssim 1$ GeV, they can be directly produced by the decay of mesons. In this case, the dark sector can be probed directly by meson decays with invisible final-state particles, and also by beam-dump experiments where the mediators are produced on-shell by the decay of mesons, see e.g. [71] for the scalar mediator case. The models then suffer from severe constraints and we do not explore that mass region in this paper.

---

[8]Within our treatment, only DUNE can put a constraint on the couplings in the isovector case. This is partly because our treatment of the Earth attenuation is too conservative since we ignore the form factor in the computation of the Earth attenuation. However, even if we include the form factor, the derived constraints are not strong, so we do not discuss this possibility in this paper.

**Constraints on other couplings.** As shown in Eqs. (1, 2), in this paper, we assume the mediators couple only to light quarks. A rich phenomenology would of course arise if one switched on couplings of the mediators to other SM particles, see e.g. [72–75]. Here we just briefly mention that a coupling to the top quark would induce, at one-loop, a $bs\phi/a$ vertex constrained by $B \to K\phi/a(\to \text{invisible})$ [76,77], and that bottom and charm couplings would be constrained by $\Upsilon \to \gamma\phi/a(\to \text{invisible})$ and $J/\psi \to \gamma\phi/a(\to \text{invisible})$ [78].

### 4.3 Direct detection

The parameter space is constrained by conventional DD experiments. In the scalar mediator model, the dark matter can scatter with nucleons by a tree-level $\phi$ exchanging diagram. The spin-independent non-relativistic cross section is given as [71]

$$\sigma_{\text{SI}} = \frac{\mu^2}{\pi} \frac{1}{A^2} \left( Z \frac{g_{\chi\phi} y_{\phi pp}}{m_\phi^2} + (A-Z) \frac{g_{\chi\phi} y_{\phi nn}}{m_\phi^2} \right)^2, \tag{48}$$

where $\mu = (1/m_\chi + 1/m_N)^{-1}$ is the reduced mass for dark matter-nucleon scattering, $A$ and $Z$ are the mass and atomic numbers of the target particle, and $y_{\phi pp}$ and $y_{\phi nn}$ are effective $\phi$-nucleon couplings defined as

$$y_{\phi pp} = g_{u\phi} \frac{f_u^p m_p}{m_u} + g_{d\phi} \frac{f_d^p m_p}{m_d}, \qquad y_{\phi nn} = g_{u\phi} \frac{f_u^n m_n}{m_u} + g_{d\phi} \frac{f_d^n m_n}{m_d}. \tag{49}$$

We take $m_u = 2.16$ MeV, $m_d = 4.67$ MeV, $f_u^N = 0.0199$ and $f_d^N = 0.0431$ for both $N = p, n$.

In the pseudoscalar mediator model, a tree-level contribution is suppressed in the non-relativistic limit both by the spin and by the velocity as shown in Eq. (3). The dark matter can still scatter with nucleons by one-loop $a$ exchanging box diagrams [79–83]. Here we follow [83] to evaluate this contribution. The spin-independent non-relativistic cross section from the box diagrams is given as

$$\sigma_{\text{SI}} = \frac{1}{\pi} \left( \frac{m_\chi m_N}{m_\chi + m_N} \right)^2 |C_N|^2,$$
$$C_N = m_N \sum_{q=u,d} \left( C_q f_q^N + \frac{3}{4} (m_\chi C_q^{(1)} + m_\chi^2 C_q^{(2)})(q^N(2) + \bar{q}^N(2)) \right). \tag{50}$$

Here $C_q$ is the Wilson coefficient for a twist-0 operator, and $C_q^{(1)}$ and $C_q^{(2)}$ are the Wilson coefficient for twist-2 operators. They are given as

$$C_q = -\frac{m_\chi}{(4\pi)^2} \frac{g_{qa}^2 g_{\chi a}^2}{m_a^2} [G(m_\chi^2, 0, m_a^2) - G(m_\chi^2, m_a^2, 0)], \tag{51}$$

$$C_q^{(1)} = -\frac{8}{(4\pi)^2} \frac{g_{qa}^2 g_{\chi a}^2}{m_a^2} [X_{001}(m_\chi^2, m_\chi^2, 0, m_a^2) - X_{001}(m_\chi^2, m_\chi^2, m_a^2, 0)], \tag{52}$$

$$C_q^{(2)} = -\frac{4m_\chi}{(4\pi)^2} \frac{g_{qa}^2 g_{\chi a}^2}{m_a^2} [X_{111}(m_\chi^2, m_\chi^2, 0, m_a^2) - X_{111}(m_\chi^2, m_\chi^2, m_a^2, 0)], \tag{53}$$

where $G$, $X_{001}$, and $X_{111}$ are loop functions defined as

$$X_{001}(p^2, M^2, m_1^2, m_2^2) = \int_0^1 dx \int_0^{1-x} dy \frac{(1/2)x(1-x-y)}{M^2 x + m_1^2 y + m_2^2(1-x-y) - p^2 x(1-x)}, \tag{54}$$

$$X_{111}(p^2, M^2, m_1^2, m_2^2) = \int_0^1 dx \int_0^{1-x} dy \frac{-x^3(1-x-y)}{[M^2 x + m_1^2 y + m_2^2(1-x-y) - p^2 x(1-x)]^2}, \tag{55}$$

$$G(m_\chi^2, m_1^2, m_2^2) = 6X_{001}(m_\chi^2, m_\chi^2, m_1^2, m_2^2) + m_\chi^2 X_{111}(m_\chi^2, m_\chi^2, m_1^2, m_2^2). \tag{56}$$

The quantities $q^N(2)$ and $\bar{q}^N(2)$ are the second moments for quark distribution functions for nucleons whose values we extract from [83].

We combine the results from NEWS-G [84], DarkSide-50 [85], CDMSlite [86], CRESST-III [87], XENON1T (S2 only) [88], and XENON1T (S1+S2) [4], and show the DD bound in Figures. 4 and 5. In the case of our interest, CRESST-III, DarkSide-50 and XENON1T give the strongest constraints in the relatively small, middle and large dark matter mass ranges, respectively.

## 5 Theoretical Consistency

The quark interactions in Eqs. (1) and (2) are effective descriptions in the phase where the electroweak symmetry is broken. A possibility to UV-complete the interactions with couplings $g_{u\phi}$ and $g_{ua}$ is by adding to the model two left-handed Weyl fermions $U_L$ and $U_R^\dagger$, with $U_{L,R}$ in the same SM gauge representations of the SM field $u_R$. We then add to the SM Lagrangian (in Weyl notation, with $H$ the SM Higgs doublet)

$$\mathcal{L}_{uU}^{\text{UV}} = y_U q_L U_R^\dagger H + M_U U_L U_R^\dagger + \text{h.c.}, \tag{57}$$

and, depending on the mediator under consideration,

$$\mathcal{L}_\phi^{\text{UV}} = g_{U\phi}\phi U_L u_R^\dagger + \text{h.c.}, \tag{58}$$

$$\mathcal{L}_a^{\text{UV}} = g_{Ua} i a U_L u_R^\dagger + \text{h.c.}. \tag{59}$$

Upon the Higgs taking VEV, Eq. (57) induces a mass mixing between $u_L$ and $U_L$ which, via Eqs. (58), (59), gives

$$g_{u\phi} = g_{U\phi}\frac{y_U v}{\sqrt{2}\,M_U'}, \qquad g_{ua} = g_{Ua}\frac{y_U v}{\sqrt{2}\,M_U'}, \tag{60}$$

where $v \simeq 246\,\text{GeV}$ and $M_U' = \sqrt{y_U^2 v^2/2 + M_U^2}$ is the physical mass of the Dirac fermion $(U_L', U_R)$ (mass eigenstate up to relative orders $y_u v/M_U'$, with $y_u$ the SM Yukawa coupling). LHC limits on new vector-like quarks constrain $M_U' \gtrsim 1.3\,\text{TeV}$ [89,90], so that

$$g_{u\phi,ua} \lesssim 0.14\, g_{U\phi,Ua} y_U. \tag{61}$$

As our new limits and sensitivities probe values of $g_{u\phi,ua}$ smaller than 0.1, they test a region compatible with a fully perturbative UV completion (i.e. with $g_{U\phi}$ and $y_U$ smaller than one). The UV completion of the couplings $g_{d\phi}$, $g_{da}$, $g_{s\phi}$, $g_{sa}$ is entirely analogous.

## 6 Conclusions and Outlook

The quest for new experimental tests of sub-GeV Dark Matter has been a booming field in recent years. One such test [7,8] consists in relying on the DM sub-component that is upscattered by cosmic rays, and that thus possesses a larger kinetic energy than DM in the virialised halo. The existence of this component, which is unavoidable as long as DM possesses any interaction with the SM, opens the possibility to test sub-GeV DM with 'standard' DD experiments like XENON1T [7], and with large neutrino detectors built for other purposes, like Super-K [8].

Within this context and focusing on DM interactions with nucleons, in this paper we obtained the following new results:

⬦ by the use of Super-K data on protons detected via their Cherenkov emission [15,16], we derived new limits on DM-nucleon cross sections that are stronger than all previous ones relying on CR-upscattered DM [7,12]. We also estimated related sensitivities at Hyper-K and DUNE;

⬦ we casted our limits and sensitivities on the parameter space of two simplified models, where DM is a fermion and its interactions with quarks are mediated by a new scalar or pseudoscalar particle, cf. Eqs. (1) and (2). This allowed us to put on solid grounds our proposal to test light DM at large neutrino experiments, because

    i) we took into account the energy dependence of the DM scattering cross sections with nuclei, which is necessary whenever the energies relevant for different detectors are different (as it is the case with Super-K vs DUNE vs XENON1T);

   ii) we compared our new limits and sensitivities with other tests of these models, like 'standard' DD, BBN, monojet and meson decays, see Sec. 4. Note that the last two tests require to specify the mediator of DM interactions and its mass. As a byproduct of this comparison, we recasted monojet searches to DM models with scalar and pseudoscalar mediators, and we assessed the status of limits on sub-GeV pseudoscalar coupling to quarks.

  iii) we assessed the theoretical consistency of the parameter space we test, by explicitly building a UV completion that is free from other constraints, see Sec 5.

Our main results are shown in Fig. 4 in the parameter space of the simplified models. For the reader's convenience, in Fig. 5 we translated them to the usual plane of DM mass vs spin-independent non-relativistic cross section. We stress that the use of our limits and sensitivities in that plane, together with others that assume a cross section constant in the energies relevant for other detection techniques, is not only consistent, but stands on very solid footing, see Sec. 3.4 for more details.

Our results open several future directions of exploration. First and foremost, they motivate the realisation of our proposed searches at Super-K, Hyper-K and DUNE. They also pave the way to study the impact of these searches, and their interplay with other experimental probes, in other simplified models of hadrophilic DM. Finally, it would be interesting to study the case where DM couples to both leptons and hadrons. We plan to come back to some of these directions in future work.

## Acknowledgements

FS thanks Benjamin Fuks for help in making MadAnalysis5 and Delphes talk to each other.

**Funding and research infrastructure acknowledgements:**

∗ Y.E. and R.S. are partially supported by the Deutsche Forschungsgemeinschaft under Germany's Excellence Strategy – EXC 2121 "Quantum Universe" – 390833306;

∗ F.S. is supported in part by a grant 'Tremplin nouveaux entrants et nouvelles entrantes de la FSI'.

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
