# Peer review of "Neutrino experiments probe hadrophilic light dark matter"

_SciPost Physics, doi:SciPost Phys. 10, 072 (2021)_

## Round 1 · Referee Report · Anonymous (Referee 1) · 2021-2-14

Report

The authors study the up-scattering of cosmic rays (CR) on MeV-GeV-scale dark matter (DM) and the subsequent re-scattering of DM on nuclei in large neutrino detectors. This is an interesting possibility because 1) it is expected to occur as long as dark matter has a substantial coupling to nucleons, and 2) it can provide interesting sensitivity to dark matter masses at the GeV scale and below. This mechanism therefore provides a complementary probe to traditional direct detection experiments, which rely on nuclear recoils induced by non-relativistic halo dark matter scattering.

This work goes beyond the existing literature in two ways. First, new strong limits are derived using Super-K data on protons detected via Cherenkov light, and future sensitivities for Hyper-K and DUNE are also presented. Second, their limits and sensitivities from CR-upscattered DM are compared with other limits from cosmology, direct detection, and various other terrestrial experiments within the context of two simplified DM models.

After presenting the simplified models of interest and the relevant details related to DM scattering, a careful discussion is provided detailing their procedure for estimating the CR-upscattered DM event rates at neutrino experiments. This includes their assumptions on the CR flux distributions and the relevant data and theoretical modeling used over different CR energy ranges, the DM distribution and local density, and their modeling of the DM attenuation in Earth. Their procedure for deriving the new limits from Super-K is is well-described and appears valid, as is their methodology for estimating future sensitivities. The authors then provide a detailed overview of a variety of other potential constraints on their simplified model parameter space. A simple UV completion of their low energy simplified models is also given.

This is an excellent paper. The physics is sound, new relevant limits are derived and interpreted within a more complete theoretical framework than has been done previously, and the presentation is clear. It satisfies all SciPost General Acceptance Criteria and also meets Expectations 1,3,4. It is a valuable addition to the literature on sub-GeV DM, and I wholly recommend it for publication.

---

## Round 1 · Referee Report · Anonymous (Referee 2) · 2021-2-15

Report

This is a well written and detailed discussion of constraints and sensitivities of large volume neutrino detectors to light dark matter (sub-GeV mass range). The main idea behind the constraints was already pointed out in the literature, but the authors provide a more complete description of the phenomenology and extend the discussion to simplified models of hadrophilic dark matter.

In my view, it meets the excellence criteria of the paper, however, I would like to suggest a few minor improvements that would help clarify certain elements of the study.

1) The choice of axial mass seems to be on the aggressive side (Ma = 1.3 GeV), motivated by the lattice calculation in ref 29 and the MiniBooNE measurements in ref 28. Given that the global average is smaller, Ma ~ 1 GeV, and that several issues are known to exist with this MiniBooNE measurement (e.g. lack of 2 body currents) it would be interesting to see how robust their conclusions are to the choice of Ma. The authors acknowledge that the constraints are stronger for larger Ma.

2) It would be interesting to foreshadow that the paper relies on an asymmetric population of dark matter in the model section or introduction, as otherwise the models considered are severely constrained by CMB. This also selects the scalar mediator case as the most interesting model.

3) Although the list of experiments considered by the authors is very comprehensive, it is lacking Borexino. It is not immediately clear why this large volume, liquid scintillator, and low threshold experiment wouldn't also be proving important constraints in the parameter space.

4) The pi0 -> invisible constraint seems to be severely constraining the choice of couplings at low masses. It would also be interesting to point to the latest pi0>inv constraint from NA62, which is stronger than what is stated in the text and emphasize.

  • validity: high
  • significance: good
  • originality: good
  • clarity: high
  • formatting: good
  • grammar: excellent

Author:  Yohei Ema  on 2021-03-05  [id 1287]

(in reply to Report 2 on 2021-02-15)
Category:
remark

The figures that are mentioned in the reply to this referee report are attached.

Attachment:

supplement.pdf

---

## Round 2 · Referee Report · Anonymous (Referee 2) · 2021-3-5

Strengths

Technical analysis
Use of a comprehensive list of experimental input
Novel constraints

Report

Dear editor and authors,

the authors have successfully addressed all my comments and initial concerns. I recommend the paper be published in its current version.

I have also compared v1 to v2, and verified the new changes are clear and that they indeed improve the manuscript.

---

## Round 2 · List of Changes

Dear Editor,

We thank the referees for the positive reports and for the constructive comments and suggestions, which we think allowed to improve the clarity and comprehensiveness of the paper. Please find below our response and a summary of the changes in our revised version of the manuscript.

1) We have performed again our analysis, now using the world-average value Lambda_a = 1.03 GeV instead of \Lambda_a = 1.3 GeV (which was motivated by MiniBooNE and also by recent lattice results). We find the world-average choice worsens our limits by order ten percent, see the Figures attached in the field of Reply to Report 2. We have added two comments on this in the main text, one at page 6 below eq.~(16) and one at page 14, just before the last paragraph of Section 3.3.

2) We agree with the referee and have anticipated, at page 3 just after Eq. (2), that either DM is asymmetric or CMB favours the scalar mediator model.

3) We thank the referee for the comment, we have now evaluated the Borexino constraints following 1810.10543, i.e. we took Np = 3.2e31, t = 1.282 yrs and required that the events should be smaller than 2.44 in the range 12.5 MeV < Ke < 15 MeV, corresponding to 21.6 MeV < Kp < 24.9 MeV (the latter is according to our computation of the conversion). We find the results displayed in the Figures attached in the field of Reply to Report 2: Borexino is always slightly weaker than KamLAND. We thus refrain from adding yet another line to our already busy plots, and we have added a discussion in the main text, see the new text at page 12 before the paragraph `Discussion'.

4) We thank the referee for pointing out the NA62 measurement, we were not aware of it. At page 18 we have added the appropriate reference, updated the BR limit and the resulting constraint on the couplings in Eq. (45).

We hope to have satisfactorily addressed all the comments raised by the referees and that the paper is now suitable for publication.

Sincerely,

Yohei Ema, Filippo Sala and Ryosuke Sato

---

## Editorial Decision

published